# Extension of Energy and Transport Scenario Modelling to Include a Life Cycle Perspective

**Simon Pichlmaier** [1,2,*], **Michael Kult** [1] and **Ulrich Wagner** [1,2]

1   FfE Munich, 80995 Munich, Germany; mkult@ffe.de (M.K.); uwagner@ffe.de (U.W.)
2   Chair of Energy Economy and Application Technology, TUM Department of Electrical and
    Computer Engineering, Technical University of Munich, 80333 Munich, Germany
*   Correspondence: spichlmaier@ffe.de; Tel.: +49-89-158121-41

**Abstract:** The paper outlines the methodology for the extension of the assessment of transport scenarios to include a life cycle perspective. When considering greenhouse gas emissions in the operational phase, the inclusion of the upstream chain increases emissions in conventional systems by only 17% to 19%. In transport systems that utilise a large share of electricity generated predominantly from renewable energies without direct emissions, this value can rise sharply. In the present case, up to 304%. The emissions currently associated with the production of the transport fleet correspond to 56 Mt $CO_2$e and thus 22% of total emissions. In most scenarios, however, this value decreases more slowly than the operational emissions. This increases the share of emissions caused by production. Thus, the inclusion of life cycle emissions is an important component for assessing sustainability.

**Keywords:** transport; energy systems analysis; energy system modelling; life cyle assessment





## 1. Introduction

In September 2020, China's President Xi made it clear that the country's $CO_2$ emissions will peak in 2030 at the latest. It is also planned to achieve climate neutrality by 2060. These targets, set by the nation with highest greenhouse gas (GHG) emissions in the world, show that the significance of the challenges posed by the climate crisis has now been recognised worldwide. Other countries are also expanding their nationally determined contributions. [1] Given this context it is important to analyse and identify the environmental impacts of current and future products and systems in order to pave the way for a transformation towards a sustainable society.

In the energy and transport sectors, scenarios are used to assess future developments. Unlike forecasts, scenario analyses provide insights into the effects of certain exogenously given framework conditions. One example are the methods of the energy system analysis, which are used to carry out many different studies on a national and international scope [2–6]. Since the development of the first energy system models in the 1970s, the number of tools for analysing energy systems has been steadily increasing [7]. In addition to the analysis of energy scenarios, transport scenarios are also modelled and examined with regard to their future energy consumption and the associated GHG [8–11]. In this context, the analyses of the energy consumption of the transport scenarios in combination with the investigation of the other application sectors often form the input for the energy system analysis.

In addition to systems analyses, the methodology of prospective life cycle assessment (LCA) is used for the assessment of future product systems. This assessment method enables an evaluation of a product system in terms of its entire upstream and downstream chain. The extraction of raw materials, production, required transport routes and possible disposal and recycling expenditures are included and can be examined in more detail. In the transport sector, electric vehicles have often been studied due to their greater production effort compared to the conventional alternatives [12–16]. But LCA analyses have already

been also carried out for other individual vehicles and in other areas of the transport sector as well.

The combination of scenario analysis in energy and transport and LCA is a recent development. Various publications have already shown and investigated the integration of LCA in the assessment of systems [17–21]. These describe fundamental principles and challenges in the integrated consideration of energy system analysis and LCA. Among other things, the problem of double counting emissions in the assessment of systems is described and solutions are outlined. Double-counting describes the problem that a process to be evaluated already exists in the upstream chain. In this case, for example, this could be a transport service. An important aspect in the prospective analysis up to 2050 is also the changes in the background data. Background data describes data that depicts the upstream chain but is not the focus of the analysis. In this instance, LCA databases are usually used. In terms of changing background data Mendoza Beltra et al. has laid the groundwork [22]. The model carculator, a python-based extension of brightway2, makes use of the change of background data to conduct prospective analyses for vehicles [23].

The previous work provides a good basis for adding a life cycle perspective to the modelling of energy and transport scenarios. However, on the one hand, the observations refer to entire national energy systems and thus focus on the import and export of emissions. On the other hand, passenger cars are analysed, but not the rest of the transport sector. The investigation of the different relevance of various modes of transport is lost. In this paper, the transport sector as a whole is examined in the context of the energy supply sector and expanded to include the life cycle perspective. The study focuses on Germany. However, the implementation would be possible with any national transport system. Thus, based on a detailed bottom-up stock-and-flow model of the German transport sector [10] and with the help of preliminary work in the literature, this paper describes how scenarios of the German transport system can be assessed in terms of LCA. In doing so, different mobility scenarios as well as transport technology scenarios and evaluated in combination with the development of the energy sector with regard to GHG emissions. The focus lies on the methodological enhancement and the interaction of the different models and scenarios as well as the evaluation with regard to possible mobility scenarios that allow a smaller number of vehicles in the fleet through intermodal mobility concepts.

## 2. Materials and Methods

The modeling approach builds on an existing transport model. In Section 2.1, the original flow of the model and the associated functions are explained before the extension of the model to include the life cycle perspective is described.

### 2.1. Model Structure

The model of the present paper is based on the framework of the transport model TraM presented in [10]. The The model in its original form allows to split the demand for total transport in passenger kilometer (pkm) and ton kilometer (tkm) among the different modalities. In this context, different usages of cars (private-owned or shared) are also understood to be different modalities. Within a certain modality (e.g., road passenger transport), different technologies (e.g., diesel car or electric car) can be used. The model describes all transport services that are mapped within the borders and thus includes road, rail, water and air transport. In the following, the model structure described in [10] is summarized. Although the present model represents transport in Germany, the calculation steps could be implemented in the same way for each transport model.

The main model input is the modal split. The modal split describes the distribution of the transport performance to the modalities and is described by the mobility scenario. Thus, by choosing the mobility scenario the user of the model can define how the passengers and goods in the scenario are transported. The second scenario selection is the technology scenario. With the choice of the technology scenario, the model decides which share of the new registration is fulfilled with a certain technology. Each technology is assigned a

capacity utilization ($\frac{pkm}{km}$ or $\frac{tkm}{km}$) and an annual mileage. In this way, the model calculates the number of vehicles of a technology to meet the transport demand. The next step is the calculation of the energy requirements. For this purpose, for each year specific energy consumptions are assigned to the respective vehicles. This specific energy consumption is then temporally resolved. In the case of electricity, this is an hourly resolution. For gases, a daily resolution is used, and for liquids, an annual resolution.

The energy system model ISAaR is used to calculate $CO_2$ emissions as well as costs associated with energy use [2]. The model determines the way different energy carriers are supplied. Depending on whether the repercussions of the development of the transport sector on the energy system are included or not, the energy system model of the energy sector is applied before or after. Since the focus in this case is on expanding the modeling to include the life cycle perspective, the investigation of the repercussions on the energy system is not carried out. Thus, the transport scenarios are each embedded in a static energy system scenario. The energy scenarios used are described in Section 2.2.3.

The results of the model TraM in its origin state address the criterias energy consumption, costs and $CO_2$ emissions. The costs include the investment costs of the vehicles, fixed operating costs and variable operating costs in the form of energy costs. The $CO_2$ emissions include only the direct emissions at the tailpipe and those generated during the provision of the energy sources. This does not include emissions from the upstream chain of primary energy sources, such as the extraction of coal for electricity production.

In order to extend this model structure to include the life cycle perspective, emissions from the production and end-of-life of vehicles must be added. Further, the energy sources are supplemented by additional upstream chain inventory components such as the supply of primary energy carrier. The previously reported $CO_2$ emissions are extended by additional GHG emissions. The unit dispatch for the scenario is still done with the energy system model ISAaR [2] described above. Analogously, an economic optimization is used to decide how the energy carriers electricity, gas, hydrogen and liquid hydrocarbons are supplied. Which emission factors apply for the individual energy conversion technologies in the different years is determined on the basis of a global energy scenario. Accordingly, it is assumed that the production of the various technologies is not limited to the country of the operation phase, which can also be inferred from the underlying data in the ecoinvent database [24]. Thus, emission factors for the mentioned energy sources can be derived for the different years. Analogous to the production of plants for energy supply, it is also assumed that the production and end-of-life disposal of vehicles is affected by global energy production. Thus, these scenarios are also included at this point. With the knowledge of the vehicle commissioning and decommissioning as well as their respective specific energy consumption and annual mileage given by the transport fleet structure, the emissions of the operating phase as well as the upstream and downstream emissions of the vehicles can finally be calculated. An overall scheme of the procedure can be seen in Figure 1.

As described, the models ISAaR and TraM determine the composition of the energy supply and transport sectors. For the implementation of the life cycle perspective, brightway2 was used [25]. The adjustments of the background scenarios described by Mendoza et al. [22] form the basis for the integration of the global energy scenarios. The background scenarios adjust the background LCA data for the different years. The background data must be used to integrate the upstream and downstream chains of the product system under consideration into the evaluation. Furthermore, the extensions of the model by Steubing et al. [26] allow the direct integration of the adjustments of the energy scenarios for future years in the Activity Browser of brightway2 [27]. The so-called superstructure approach is used to adapt the underlying LCA database ecoinvent 3.7 for future years using the global scenarios from [28]. In the following, the effects of the upstream chain and the downstream chain are summarised. Thus, the term production in the results includes both production and recycling.

In addition, for the transport sector the extensions carculator [23] and carculator-truck [29] exist, containing data for the development of road vehicles. These are used for

the calculations of the upstream and downstream emissions of the vehicles but not for the specific consumption and annual mileage, since these can already be derived from the transport model TraM.

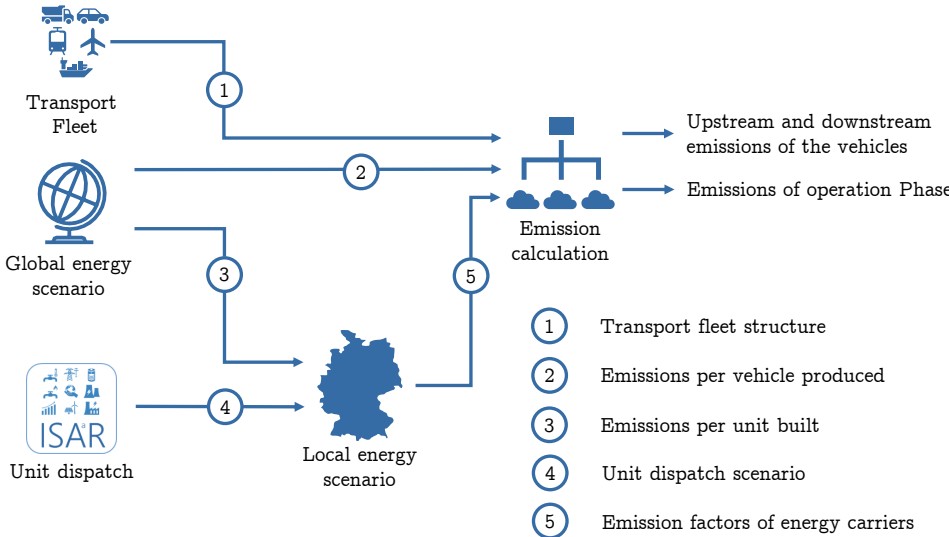

**Figure 1.** Basic scheme of the used methodology for the extended model.

### 2.2. Scenarios

When selecting the scenarios to be evaluated, a distinction is made between mobility scenarios, local (here Germany) technology and energy scenarios, and global energy scenarios. The following section describes the characteristics of the individual scenarios. In this context, the local energy scenario and the global energy scenario have to be synchronized. It is thus assumed that efforts to achieve climate targets in the country which is analysed will only take place if global efforts are also made at the same time. Furthermore, ambitious technology scenarios in the transport sector are only combined with ambitious energy scenarios. At the end of this section, the scenario combinations are defined in tabular form.

#### 2.2.1. Mobility Scenarios

A discussion of different mobility scenarios has already taken place in [30]. In addition to a reference scenario, the effects of car sharing and autonomous driving are examined on the basis of the original, non-extended model. For the present study, an intensive car sharing and multi-modal scenario (CSI) will be investigated as a comparison to the reference scenario (Ref). An overview of the development of passenger transport performances is shown in Figure 2.

In the scenario Ref, there is only a very slight change in both the total performance and the shares of modalities. Over the entire period the largest share of transport performance is provided by private cars. Slightly more than 10% of the transport performance is provided by airplanes, while the other transport modes have smaller shares. At this point, it should be noted that for airplanes, that all flights taking off from Germany are allocated to Germany's transport performance. For all other modes of transport, the transport performance within the borders is allocated to Germany in accordance with the territorial principle.

If looking at the CSI scenario, one can see that after 2025 the share of transport performance by private cars decreases as it is shifted to a combination of shared cars and public transport such as buses and trains. For this purpose, potentials for users of multi-modal transport in urban and rural areas are derived on the basis of the mobility study Mobility in Germany [31]. In the CSI scenario, these potentials are exploited to the maximum. A detailed explanation of the development of the scenario is given in [32].

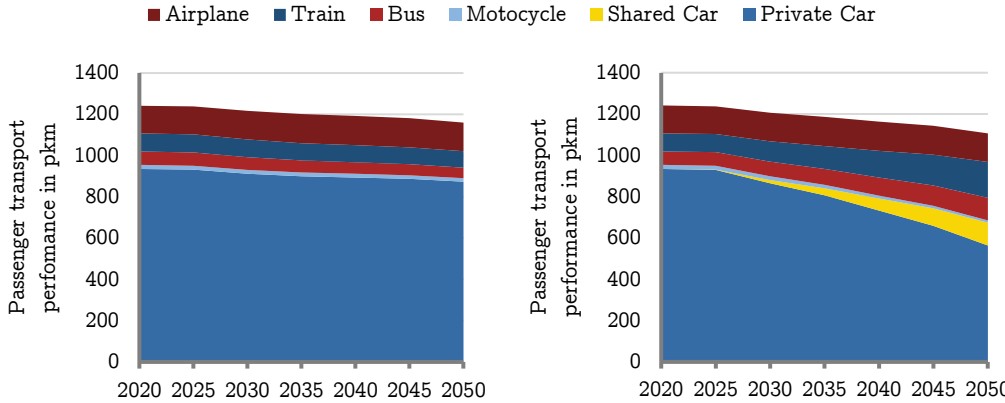

**Figure 2.** Passenger transport performance of the reference scenario Ref (**left**) and the multi-modal scenario CSI (**right**).

The development of freight transport performance stays the same in both scenarios. The scenario taken from [11] describes an increase in freight transport that is mainly driven by road transport. It is characterized by consistent share of slightly more than 80% of the total freight transport performance.

### 2.2.2. Transport Technology Scenario

The transport sector is composed of road, rail, water and air transport. As mentioned above, the regional principle applies with the exception of air transport. In the case of shipping, only domestic shipping is part of the analysis. For air traffic, all flights departing from Germany are included.

A conservative and an ambitious technology scenario for the transport sector are examined. Both scenarios were first described in the project Dynamis [33]. The conservative scenario is designed in such a way that the shares of the technologies used remain largely the same. Thus, in private car transport, for example, there will still be a large proportion of vehicles in the fleet being powered by an internal combustion engine in the year 2050 (see Figure 3, top left). In the area of freight transport, the current propulsion mix is continued for the future and thus almost exclusively diesel vehicles are used. In rail transport, no further electrification takes place and the propulsion technologies for inland waterways and air transport will remain the same. If emission reduction targets are to be achieved in this conservative scenario, the liquid hydrocarbons still required must be provided by the supply sector on a renewable basis. This is accompanied by large losses on both the application side and the energy supply side.

The conservative scenario is opposed by an ambitious scenario. In the scenario, road transport undergoes a major change in propulsion until 2050. Vehicles with combustion engines in the private car sector are gradually replaced by alternatives. In 2050, the share of battery electric vehicles (BEV) in the stock amounts to 65%. In addition, the fleet contains 11% fuel cell electric vehicles (FCEV) and 8% plug-in electric vehicles (PHEV). The remaining vehicles with combustion engines are largely powered by compressed natural gas (CNG). The change in propulsion technology in truck traffic is taking place analogously. In 2050, more than 50% of semi-trailer trucks will be battery-powered. About 20% of the vehicles will run on hydrogen and the rest will continue to run on diesel. The other areas of freight transport on the roads are also undergoing a similar development in smaller road freight segments. The railway routes on which it is economically viable will continue to be electrified with overhead lines. In addition, hydrogen is being introduced into rail transport. Parts of inland waterway transport will be converted to hydrogen, but the majority will continue to run on diesel. In air transport, only kerosene will continue to be the only energy carrier used until 2050.

In addition to the above-mentioned sub-areas, parts of the passenger transport service are covered by car sharing vehicles. These already have a very high penetration of electric

vehicles and, based on the nearer future development, it is assumed that in both scenarios no more combustion engines will be used by 2050. While more than 85% of the shared fleet in 2050 is based on BEV, the remaining vehicles are composed of PHEV and FCEV.

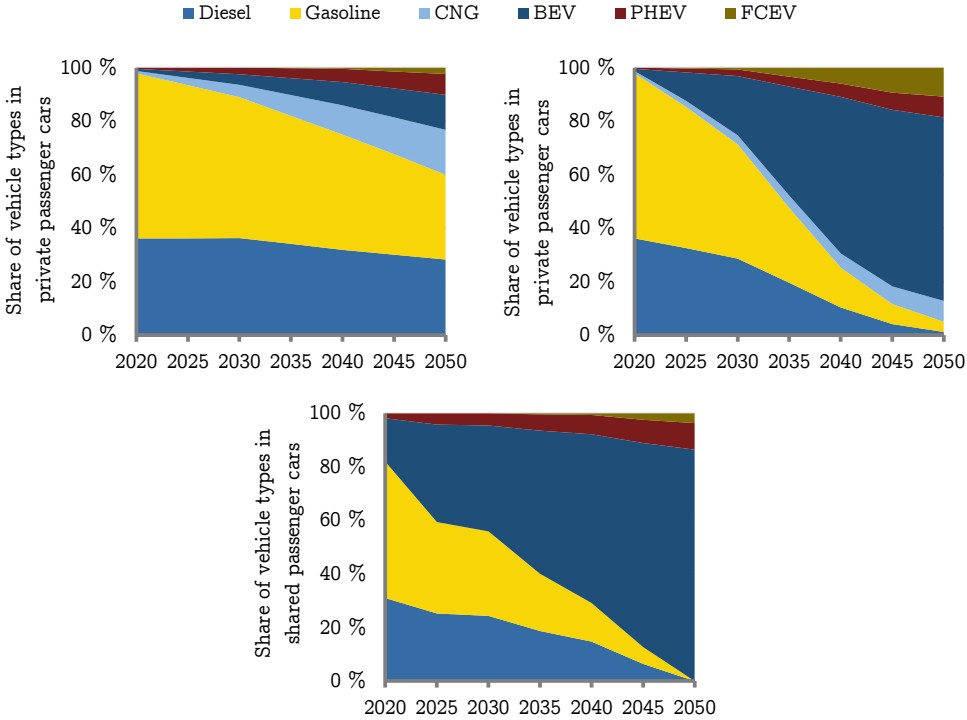

**Figure 3.** Technology split for privately owned cars in the conservative (**top left**) and the ambitious (**top right**) scenario as well as the technology split of the shared cars in both scenarios (**bottom**).

### 2.2.3. Energy Scenarios

When talking about energy scenarios for prospective LCA analyses in the German transport sector, one has to distinguish between local and global scenarios. On the one hand, large shares of the energy consumption in the production of vehicles and their components are distributed across different countries. In addition, energy generation technologies such as photovoltaic modules are largely produced abroad. A consideration of the global development in the area of energy supply is therefore necessary. On the other hand, energy carriers are consumed in Germany in the operating phase of the vehicles. Accordingly, the focus in the operating phase lies on German energy supply.

The global energy scenarios, on the contrary, are included in the calculation as a background life cycle inventory (LCI). The background data is based on ecoinvent 3.7 [24] and is adjusted for different years as described in Section 2.1. Fur this purpose, two different scenarios are used. The Base scenario corresponds to a business as usual development. In this scenario, global GHG emissions rise sharply until 2070 before the first decrease. The second scenario ClimPol (Climate Policy) reaches the peak of its GHG emissions between 2015 and 2020. After 2020, emissions fall significantly. The scenario corresponds to the goals of the Paris Climate Agreement to limit global warming to well below 2° Celsius. The GHG emissions of the scenarios can be seen in Figure 4. It remains to be mentioned that the numbers in the figure only represent the emissions without GHG sinks such such as forests or moors.

With regard to the background processes for the German transport sector, in particular the development in China (CHA) is of great interest. Since this is where large parts of the value chain for electric vehicles are currently located and will continue to be located in the future. Using LCA methodology, these emissions are assigned to the product, in this case the transport system in Germany.

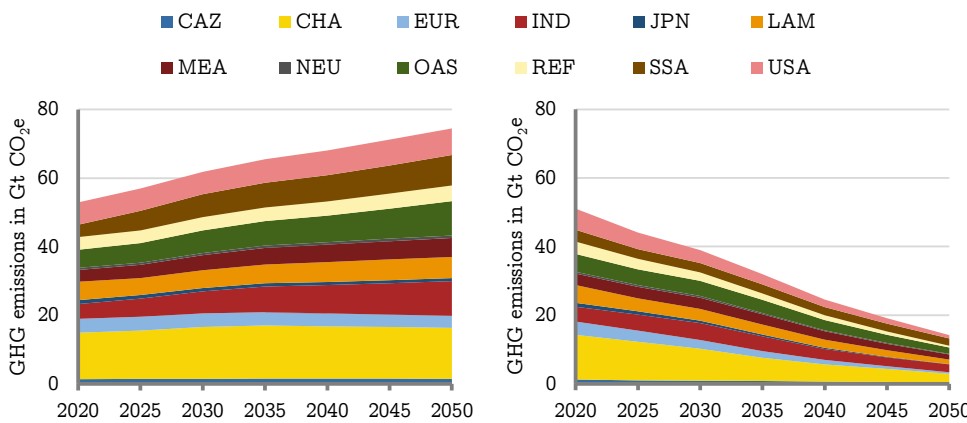

**Figure 4.** GHG emissions of the global energy scenarios Base (**left**) and ClimPol (**right**); own illustration based on [22,28].

In consistency with the international scenarios, a conservative and an ambitious energy scenario for Germany are used. These scenarios were developed and described in the Dynamis project [33]. Based on the global energy scenarios, the terms Base and ClimPol are also used for these two scenarios. For Germany, as well, the Base scenario depicts a future development in which the climate protection goals of the Paris Agreement will not be achieved. According to the current legal framework, a phase-out of nuclear energy will be completed in 2022 and a phase-out of coal-fired power generation in 2038. By 2050, the electricity system is largely based on renewable energy and gas combustion. No renewably produced hydrocarbons are used and hydrogen is still obtained from steam reforming. This is in contrast to the ClimPol scenario. Here, despite a strong increase in electricity demand, more than 90% of German electricity production will be realised by renewable energies by 2050. Hydrogen is provided entirely by electrolysis in 2050 and fossil-based hydrocarbons are replaced by renewable energy-based alternatives. Through carbon capture and storage (CCS), the remaining $CO_2$ emissions are captured and stored [33].

In the project Dynamis, emission factors were developed for the energy carriers electricity, methane, hydrogen and liquid hydrocarbons based on their supply. However, these only included energy-related $CO_2$ emissions. For this analysis, these are expanded to include their entire upstream chain as well as other GHG emissions apart from $CO_2$. The resulting emission factors are listed in Table 1.

**Table 1.** LCA emission factors in kg $CO_2$e/kWh in Germany derived from the two scenarios of the project Dynamis [33] and extended by the upstream chain and other GHG emissions.

| Scenario | Energy Carrier | 2020 | 2030 | 2040 | 2050 |
|---|---|---|---|---|---|
| Base | Electricity | 0.45 | 0.35 | 0.18 | 0.15 |
| Base | Methane | 0.25 | 0.25 | 0.25 | 0.25 |
| Base | Hydrogen | 0.31 | 0.31 | 0.31 | 0.31 |
| Base | Liquid Hydrocarbons | 0.30 | 0.30 | 0.30 | 0.30 |
| ClimPol | Electricity | 0.45 | 0.28 | 0.12 | 0.05 |
| ClimPol | Methane | 0.25 | 0.25 | 0.25 | 0.16 |
| ClimPol | Hydrogen | 0.31 | 0.32 | 0.28 | 0.06 |
| ClimPol | Liquid Hydrocarbons | 0.30 | 0.30 | 0.30 | 0.04 |

In contrast to the consideration of only energy-related $CO_2$ emissions the application of LCA methodology leads to residual emissions in 2050 even in the ClimPro scenario resulting from the production of energy sources. This is due to the fact that although zero emissions are realised in Germany in ClimPol, GHG emissions enter the system through upstream chains of energy carriers from other countries. This is, for example, the case for

the production of photovoltaic systems. A complete reduction of emissions would at this point only be possible if all regions of the world from which raw materials and products are sourced had already reduced their emissions to zero in 2050. As it can be seen in Figure 4, this is not the case.

Finally, Table 2 lists the scenarios that are examined in the present analysis. As already mentioned, conservative technology scenarios are only combined with conservative energy scenarios. This applies analogously to ambitious scenarios.

**Table 2.** Combination of scenarios.

| Name | Mobility Scenario | Technology Scenario | Energy Scenario |
|---|---|---|---|
| Ref-Base | Ref | conservative | Base |
| Ref-ClimPol | Ref | ambitious | ClimPol |
| CSI-Base | CSI | conservative | Base |
| CSI-ClimPol | CSI | ambitious | ClimPol |

## 3. Results

The results first deal with the changes due to the extension in the form of a comparison of both methods with and without the inclusion of the upstream GHG emissions. Subsequently, the different scenarios are compared in terms of their total GHG emissions. First, the focus is placed on the share of the fleet's production in total emissions. This is followed by a closer look at the emissions with regard to the passenger car fleet.

### 3.1. Comparison of Methods

The results relating to emissions in the operational phase are used to compare the two methods, as these are the only emissions accounted for by the original method. This also provides an insight into which energy sources are responsible for the largest shares of emissions. For this purpose, the resulting GHG emissions caused by the operational phase in the scenarios Ref-Base and Ref-ClimPol are plotted in Figure 5. In addition, the diagram includes the increase in emissions due to the integration of upstream chain emissions in the provision of energy sources in comparison to the evaluation without considering the provision of primary energy carriers as well as the production of the energy conversion technologies such as e.g., photovoltaic modules.

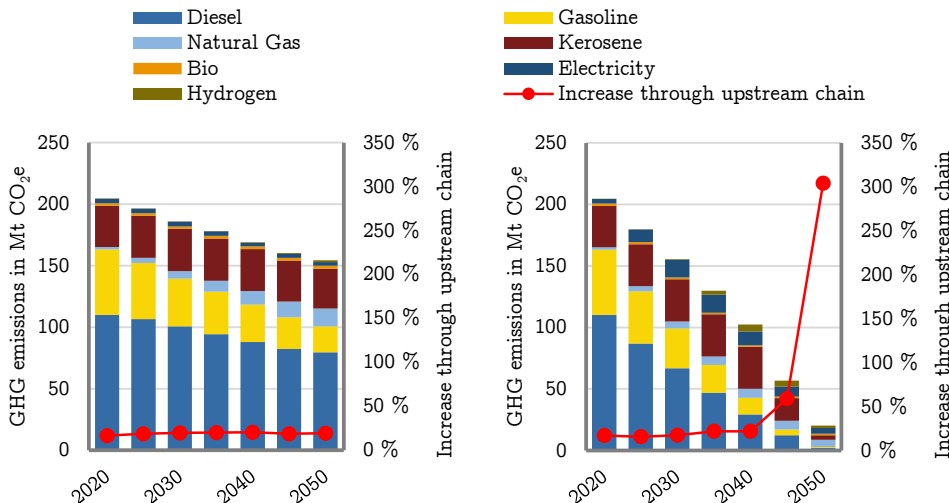

**Figure 5.** Development of emissions caused by energy demand in the operating phase in the scenarios Ref-Base (**left**) and Ref-ClimPol (**right**); increase through the inclusion of the GHG emissions arising from the provision of primary energy carriers and the production of energy conversion technologies.

First of all, it becomes clear that the use of internal combustion engines in the conservative Ref-Base scenario will continue the use large amounts of liquid hydrocarbons until 2050 and thus lead to emissions. Emissions fall by 24.5% from 204 Mt in 2020 to 154 Mt $CO_2e$ in 2050. On the contrary, emissions are drastically reduced in the Ref-ClimPol target scenario by 2050. On the one hand, this is related to the use of more efficient technologies such as BEVs and FCEVs in various road transport modes. In addition, the provision of energy sources in Germany changes towards more renewable energies. Finally, the upstream chain will also become less emission-intensive due to the decreasing emissions associated with the provision of raw materials and the production of energy conversion technologies. In the time horizon considered, the emissions in this scenario decline from 204 Mt to 20 Mt $CO_2e$. This implies a reduction of 90.1%.

Figure 5 also shows the proportion by which emissions are increased by including the upstream chain. The original method does not include the provision of primary energy carriers and the construction of plants for energy conversion. Adding these two parameters results in an increase in emissions of 17% for the year 2020 when analysing the operational phase. This is essentially due to the provision of coal, oil and gas. When considering the conservative scenario, this value remains constantly below 20% until 2050. Examining the ambitious scenario, it becomes apparent that the increase rises sharply due to the inclusion of upstream chains in 2045 and 2050. This is due to the increasing shift towards renewable energies, since they have no direct emissions and thus only upstream chain emissions take effect.

### 3.2. Comparison of Scenarios

Now a deeper look is taken at the evaluation of the scenarios. In Figure 6 the total emissions for the different scenarios are depicted. In 2020, the total emissions of the German transport sector, including the upstream chains of the energy sources and the emissions from vehicle production, are 261 Mt CO2e. Of these, 22% are attributable to vehicle production. The production includes cars and trucks as well as trains, ships, aircrafts and other modes of the German transport system.

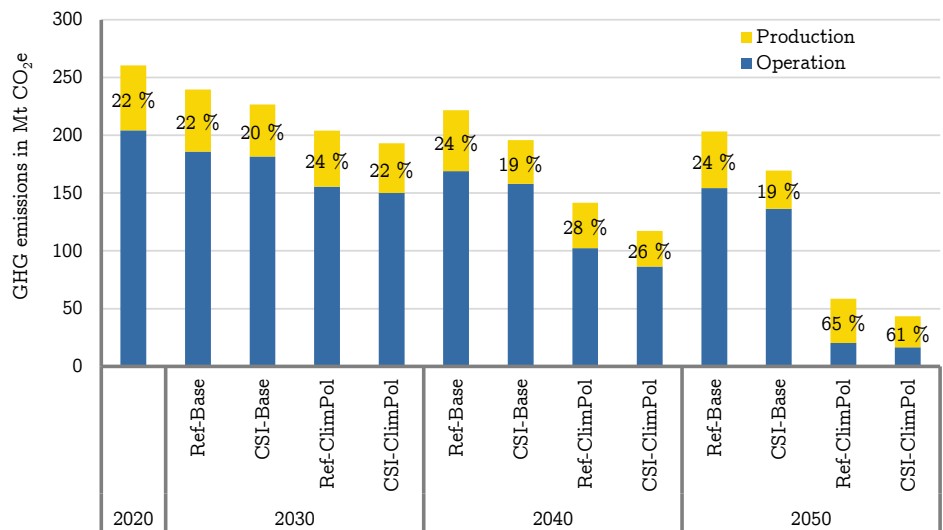

**Figure 6.** Developments in GHG emissions from production and operation in all scenarios.

When looking at the different scenarios, it becomes evident that the emissions in the conservative Base scenarios decrease more slowly than those in the ambitious ClimPol scenarios. In the two Base scenarios, the share of emissions from vehicle production is lower than in the ambitious scenarios, with a value of 24% and 19% for Ref-Base and CSI-Base respectively. In the ambitious scenarios, a sharp reduction in emissions is possible due to the strong drop in operating emissions. With 65% and 61% the share of emissions

in production is much higher in 2050 than in the previous years. This is due to the fact that emissions from the provision of energy sources in Germany are falling faster than global emissions for vehicle production. Furthermore, it becomes apparent that the share of emissions from vehicle production and thus the total emissions are lower in the car sharing scenarios (CSI) than in the scenarios with conventional transport development (Ref).

In all scenarios and all years, the production of the passenger car fleet accounts for between 60% to 80% of the emissions originating from the production of the total transport fleet. The lowest values are reached in the car sharing scenarios in 2050, because the CSI scenarios are characterized by the most significant reduction of cars in the system. Since the share of the passenger cars in total emissions is significant, the origin of emissions from the passenger car fleet is discussed in more detail below.

Figure 7 shows the development of emissions from the production of passenger cars in the four scenarios. It can be seen that in all scenarios total emissions decrease steadily. The majority of the emissions of the car fleet come from glider production. The glider corresponds to the vehicle without a powertrain. Other noteworthy emissions are generated in the production of the powertrain as well as battery storage systems. While the respective shares of glider and powertrain production remain roughly the same relative to each other, the share of battery storage increases in the ambitious scenarios until 2030. This is primarily due to the increasing share of electric vehicles in the transport systems. However, it also becomes evident that the absolute emissions from battery production decrease after 2030. This can be explained by the fact that although the number of electric vehicles in the fleet is increasing, the specific emissions from battery production are decreasing significantly. The second effect overcompensates the first at this point. The specific emissions from the production of the glider of the various drive technologies do not differ greatly. Thus, the absolute emissions attributable to the glider are essentially related to the absolute number of passenger cars. The total passenger car fleet with private and shared vehicles contains 42.3 million cars in the Ref scenarios in the year 2050. The CSI scenario, on the other hand, with a strong trend towards car sharing embedded in an intermodal transport system, requires a total of 25.2 million vehicles in 2050. Compared to their Ref scenarios, the CSI scenarios result in a reduction of emissions from the production of the glider of 54.7% and 58.5%, respectively. The effects in the comparison between the two mobility scenarios Ref and CSI are thus more evident than those of the two energy and technology scenarios Base and ClimPol.

The specific emissions from the production of the glider of the various drive technologies do not differ greatly. Thus, the absolute emissions attributable to the glider are essentially related to the absolute number of passenger cars. The total passenger car fleet with private and shared vehicles contains 42.3 million cars in the scenarios Ref in the year 2050. The CSI scenario, on the other hand, with a strong trend towards car sharing embedded in an intermodal transport system, requires a total of 25.2 million vehicles in 2050. Compared to their conventional counterparts, the scenarios CSI result in a reduction of emissions from the production of the glider of 54.7% and 58.5%, respectively. The effects in the comparison between the two mobility scenarios Ref and CSI are thus more evident than those of the two energy and technology scenarios Base and ClimPol.

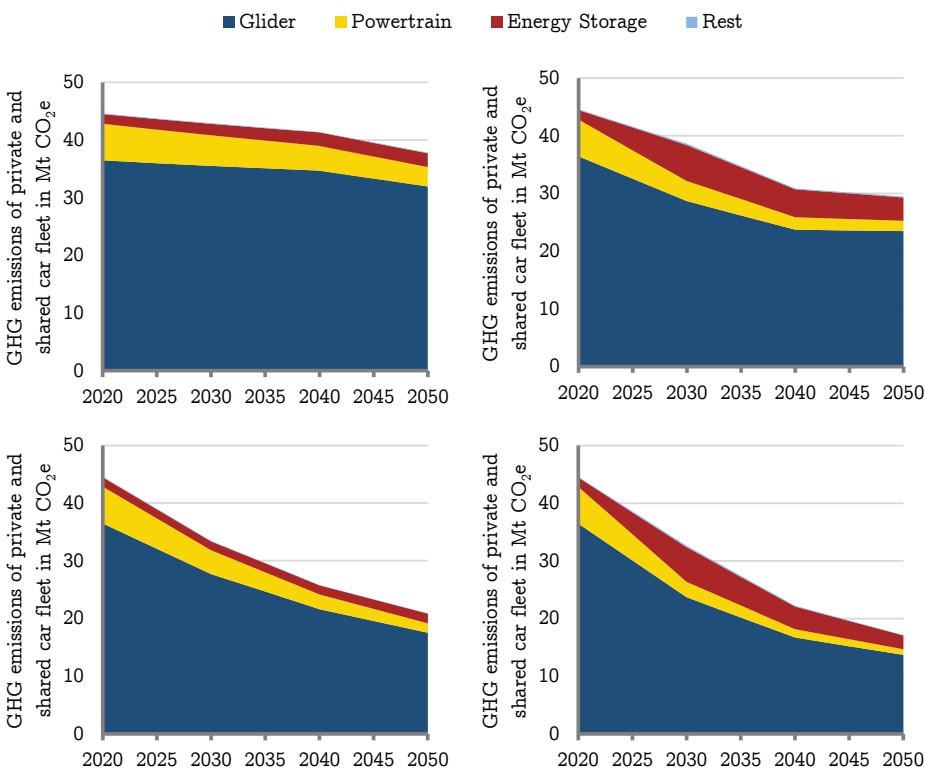

**Figure 7.** Developments in GHG emissions from the production of the car fleet in the scenarios Ref-Base (**top left**), Ref-ClimPol (**right top**), CSI-Base (**left bottom**) and CSI-ClimPol (**right bottom**).

## 4. Discussion

The extension of the modelling of scenarios in the transport sector to include the LCA gives the evaluation another important perspective. Especially in scenarios in which fewer conventional energy producers are used and the focus is on renewable energies. Nevertheless, it is still important to use different methodologies and not to perceive the LCA as the only correct method at this point. Especially the prospective LCA is based on many assumptions that can also be decisive for the result. A deep understanding of the scenarios as well as of the emerging effects is therefore essential and a classification of the results must be carried out. If the method is applied to larger regions or other energy application sectors, there is a risk of overestimating the emissions through double-counting. In the present case of the German transport sector, double-counting of emissions also occurs, e.g., due to transport services in the upstream chain of energy sources that take place in Germany. However, this effect is estimated to be very low.

The emissions in relation to production compared to the respective operating emissions are very robust and provide an estimate of the magnitude of the contribution of vehicle production to transport sector emissions. Nevertheless, the scenarios themselves are heavily laden with assumptions. For example, car sharing was integrated into an intermodal transport system. In terms of reducing emissions, this is the optimal case. However, a strong trend towards car sharing might also replace cycling or public transport routes. Thus, it would only cause more car trips and would therefore not be conducive to emission reduction.

The emissions generated in the production of the vehicles also depend on many parameters. On the one hand, there is the question of how renewable energy will develop worldwide. On the other hand, the technologies that will be used in the future, e.g., in battery storage systems for electric vehicles, are not yet clear. Finally, the question of where components will be produced is also fraught with uncertainty. Sensitivity anal-

yses would be necessary here to shed more light on the effects that arise and to derive suitable conclusions.

## 5. Conclusions

Despite the limitations described in Section 4, clear statements can be derived from the analysis. On the one hand, a broadening of perspective through the LCA in strongly renewable energy systems is very informative for the evaluation of energy carriers. By expanding the method to include the upstream chain of energy sources, emissions in the conventional case only increase by approx. 19%. However, when considering ambitious energy scenarios, the emissions from energy supply can far exceed those from energy use, as in the present case. This is mainly due to the use of photovoltaics and wind, which do not produce any direct emissions. This lead to an increase of the emissions in the operation phase of the transport of 304%.

From an LCA perspective, 22% of GHG emissions from the transport sector are attributable to vehicle production in 2020. This corresponds to 56 Mt $CO_2$e in absolute terms. In all scenarios, this value decreases steadily. With the exception of the most conservative scenario Ref-Base, the decrease is slower than in the operational emissions. Even if the production emissions represent only a minor part of the current total emissions, it is important to keep an overview of the emissions in the upstream chain. Especially car manufacturers can consider sustainability in the supply chain in this regard. With the establishment of a European battery production for electric vehicles, there is also the opportunity to establish new sustainability standards.

**Author Contributions:** Conceptualization, methodology, formal analysis, writing—original draft preparation, S.P.; software, validation, data curation, M.K. and S.P.; writing—review and editing, M.K. and U.W.; supervision, U.W. All authors have read and agreed to the published version of the manuscript.

**Funding:** This research was conducted as part of the BEniVer project, which is supported by the German Federal Ministry for Economic Affairs and Energy under grant no. 03EIV116C. The responsibility for the contents lies solely with the authors.

**Institutional Review Board Statement:** Not applicable.

**Informed Consent Statement:** Not applicable.

**Acknowledgments:** Special thanks go to Sofia Haas, Anika Neitz-Regett and Stephan Kigle, who made this paper possible through fruitful discussions as well as their work on input data, model implementations or administrative support.

**Conflicts of Interest:** The authors declare no conflict of interest.

## Abbreviations

The following abbreviations are used in this manuscript:

| | |
|---|---|
| LCA | Life cycle assessment |
| pkm | Passenger kilometer |
| tkm | Ton kilometer |
| GHG | Greenhouse Gas |
| LCI | Life cycle inventory |
| CCS | Carbon capture and storage |
| BEV | Battery electric vehicle |
| PHEV | Plug-in electric vehicle |
| FCEV | Fuel cell electric vehicle |
| CNG | Compressed natural gas |

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
