# Peer review of "Extension of Energy and Transport Scenario Modelling to Include a Life Cycle Perspective"

_futuretransp, doi:10.3390/futuretransp1020012_

Round 1

Reviewer 1 Report

The paper analyses the energy scenario of transport systems including life cycle perspectives.

General observations

Although the paper is very current and of great interest, it is necessary to make some changes to make it suitable for publication.

First of all, it is necessary to extend the literary review. Indeed, although the analysed topic is widely discussed in the literature, the state-of-the-art in the paper is limited to only 28 contributions. These are very recent (most of them are between 2017-2021) but perhaps some less recent but relevant (and historical) element is missing in the multi-objective analyses or in the overall energy analyses.

The authors should highlight the innovative contribution of their proposal with respect to the existing literature, the gap filled by means of their proposal and main limits/research prospects of their study.

Then, although the topic treated is very current and of great interest to any nation in the world, the current structure of the paper is German-centric, making the contribution very limited. So, I suggest to modify the paper framework in order to make it suitable for the publication as applicable in other national contexts.

For example, the first paragraph (page 1, line 13, introduction) the authors begin by referring to the German federal court. It would have been more useful to begin the paper by describing the growing worldwide interest in energy issues and citing the German federal court exclusively to corroborate this assertation.

Obviously, what I suggested for the first lines of the contribution should be applied by the authors to the whole paper.

Finally, some linguistic revisions may be necessary to avoid problems such as the use of German term "Benzin".

Minor observations

At page 4, line 158. The text “In Ref, there is only …” seems that there is a missing number citation.

At page 6, figure 3. The yellow fuel is indicated as “benzin”. I reckon that the correct English name should be petrol [British English] or gas/gasoline [American English]

Author Response

Thank you very much for your review, it helped me a lot to improve the paper in the areas I mentioned. You will find the updated manuscript with tracked changes attached. The following changes were made in response to your comments:

  • Changed "Benzin" to "Gasoline" in Figure 3
  • I did another broader literature review and added some older sources. However, as the coupling of LCA and energy system analysis is a newer concept, the sources are not older at this point.
  • In addition, following the literature review, I have further clarified the research gap that the paper deals with.
  • In developing the model structure, I have taken the focus away from Germany and generalised the argument. In the case of the scenarios, this is not possible any further, as this depends heavily on the availability of data. Nevertheless, the framework of the paper has been adapted to the extent that the model structure has become more international.
  • The introduction and other text passages have been adapted and internationalised.
  • In line 158 I added the word "scenario", no reference number was missing here.

Reviewer 2 Report

This study analyzes the methodological extension of the energy and transport system analysis and concluded the inclusion of life cycle emissions is an important component for assessing sustainability. The model is properly formulated and the results are tested using numerical analysis.

Author Response

Thank you for your review!

After incorporating the reviewers' comments, some minor changes have been made. You will find them in the attachment.

Reviewer 3 Report

General remark

The article deals with very important issues related to climate protection. Transport is one of the major sources of greenhouse gas emissions (GHG). The life cycle assessment (LCA) approach used to determine total GHG emissions is perfectly appropriate. LCA covers costs and emissions from the production, operating and recycling phases. In fig. 6, the recycling phase is omitted. In the case of vehicles with conventional powertrain, this phase is insignificant and may be omitted. For eclectic vehicles, the costs of the recycling phase are high. This applies in particular to the recycling of Li-Ion batteries. At present, the cost of producing Li-Ion batteries is lower than the costs of recycling. Today it is not known what will happen with millions of tons of used batteries. The 2015 Paris Agreement recognized that in order to keep global warming below 2 degrees Celsius, there should be at least 100 million electric vehicles on the roads by 2030. Already in 2025, 11 million tons of used batteries may appear on the market. Therefore, all scenarios with ambitious technology scenario (Table 2)  may look a bit different.

Detailed remarks

  1. What does the word "Glider" mean. According to the Cambridge Dictionary glider is “an aircraft that has long fixed wings and no engine and flies by gliding”. Is it a vehicle without a powertrain.
  2. In order to better show the changes in the total emission, it would be good to show all the GHG emission waveforms from Figure 7 on one graph. The differences between the scenarios would be more visible.

Author Response

Thank you for your review. Based on that I could definitely improve the paper and clarify some things. In combination with the comments of other reviewers, the attached document has now been created.

I responded to your points as follows:

  • Under the term "production" I have combined upstream and downstream emissions. Recycling or end-of-life is not excluded. I have now also made this clearer in the revision. 
  • Figure 7 intentionally focuses on cars in order to illustrate the reduction in GHG emissions that would be possible by reducing the number of vehicles. In contrast, the emissions from the production of means of transport, which have lower idle times and thus a high utilisation rate, are significantly lower and don’t differ a lot. I have therefore retained this evaluation.
  • I added the definition of the glider in the present case.

Round 2

Reviewer 1 Report

The current version of the paper has satisfied all my previous observations